# Incongruent Views of Functioning between Patients and Gastroenterologists: A Mixed Methods Study

**DOI:** 10.3390/healthcare11010062

**Published:** 2022-12-26

**Authors:** Francisco José Eiroa-Orosa, Alicia Georghiades, Amanda Rodriguez-Urrutia, Anna Accarino

**Affiliations:** 1Section of Personality, Evaluation and Psychological Treatment, Department of Clinical Psychology and Psychobiology, School of Psychology, University of Barcelona, 08035 Barcelona, Spain; 2Consultation-Liaison Psychiatry Unit, Department of Psychiatry, University Hospital Vall d’Hebron, CIBERSAM, 08035 Barcelona, Spain; 3Department of Psychiatry and Legal Medicine, Autonomous University of Barcelona, 08193 Barcelona, Spain; 4Digestive System Research Unit, University Hospital Vall d’Hebron, CIBEREHD, 08035 Barcelona, Spain; 5Department of Medicine, Autonomous University of Barcelona, 08193 Barcelona, Spain

**Keywords:** distress, functional digestive diagnosis, gastroenterology, illness experience, incongruence, patient-physician relationship

## Abstract

Gastrointestinal patients, especially those diagnosed with functional digestive disorders (FGIDs), usually present a complex clinical picture that poses a challenge for their management in primary care. The main objective of the current research was to examine the relationship of the congruence of the perception of severity and quality of life between gastroenterologists and their patients with psychological distress and the helping attitude experienced by the latter attended in primary care centres. Additionally, we wanted to explore patients’ and practitioners’ perceptions. We performed a cross-sectional study with a total of 2261 patients (1562 analysed) that attended three primary care centres. Patients completed questionnaires that measured physical functioning, distress, and perception of helping attitude. Gastroenterologists registered the functional status of each participating patient. Patients were then invited to take part in the qualitative part of the study if they were considered to have incongruent views on their functioning with their gastroenterologist. In total, 52 patients took part in one of eight focus groups. Additionally, four individual interviews were carried out with three gastroenterologists and one consultation-liaison psychiatrist specialised in FGIDs. Both incongruence and functional diagnosis correlated with distress. However, incongruent views between patients and gastroenterologists explained more variance. Statistically significant differences in patients’ helping attitude perception were detected between diagnostic but no incongruence groups. In the second stage of the study, a total of five themes were identified from the patient focus groups and the gastroenterologist interviews: illness-emotional and personal problems, disease-health system interaction, health system, intervention, and patients. The current research allowed us to confirm that incongruence in the perception of severity and quality of life between gastroenterologists and patients is related to psychological distress and that this occurs in a multifactorial context where the characteristics of the disorder interact with those of the health system.

## 1. Introduction

Functional gastrointestinal disorders (FGIDs) are disorders of the brain-gut interaction [1,2]. The most common, as diagnosed according to the Rome IV criteria, is irritable bowel syndrome (IBS), which consists of abdominal pain associated with altered bowel habits of diarrhoea, constipation, or alternating between both. Other common FGIDs include functional dyspepsia (pain or discomfort in the upper abdominal area and a feeling of fullness, bloating, or nausea), functional vomiting, functional abdominal pain, and functional constipation or diarrhoea [1].

According to the biopsychosocial model, the development of functional digestive symptoms is the result of complex and reciprocal interactions between biological, psychological, and social factors that can alter the stress response [3]. Accordingly, patients with FGIDs usually report poor health-related quality of life; higher rates of psychological distress; and other functional comorbidities, such as chronic fatigue syndrome, fibromyalgia, and chronic pain [4,5,6,7].

### 1.1. Incongruence between Clinicians’ and Patients’ Perceptions

Previous research found that incongruence (i.e., disagreement) between the patient’s and the clinician’s perception of severity and quality of life outcomes is high in conditions such as asthma [8], macular degeneration [9,10], medical comorbidities in chronic depression [11], and multiple sclerosis [12], as well as within procedures, such as hip arthroplasty [13].

Furthermore, incongruent views between clinicians’ and patients’ levels of functionality also appear to be related to higher levels of psychological distress for patients with FGIDs in tertiary care settings. These discrepancies can be linked to a communication gap between patients and gastroenterologists [14]. This may be due to a ‘mismatch’ between the patients’ beliefs about their illness and the gastroenterologists’ understanding [15]. For instance, it was found that patients’ perceptions of symptom burden are underestimated by gastroenterologists and there are differences between the patients’ and gastroenterologists´ views on the best treatment options [14]. These differences often have consequential effects on the patient-physician relationship. For instance, patients diagnosed with IBS have described physicians as being unsympathetic and hostile towards them, whereas physicians have described these patients as demanding and difficult to manage [15,16].

### 1.2. The Present Study

To our knowledge, no one has explored the influence of the incongruence between patients and gastroenterologists regarding the severity and quality of life in primary care centres where a less severe and more diverse patient population than in tertiary settings can be found. To fill this gap, the current study adopted a mixed methods approach by combining quantitative and qualitative data collection methods. The aim of the first stage of the research was to test the hypothesis that the incongruence between the perception of quality of life of patients and gastroenterologists correlates with psychological distress, adopting a similar procedure that was already conducted in tertiary care [17] but applying it to primary healthcare settings. A sequential approach was adopted (see Figure 1) to identify patients with incongruent views of their functionality with their gastroenterologists that could take part in the second stage of the study, which consisted of exploring the perceptions of patients and practitioners through focus groups and interviews, respectively.

## 2. Materials and Methods

### 2.1. Overall Study Design

The research design was developed to provide a methodological structure that could elicit in-depth information regarding the perception of severity and quality of life among patients and practitioners. A sequential approach was adopted, where the data retrieved from stage 1 were analysed to identify patients with incongruent views with physicians. These patients were then invited to participate in the second stage of the research.

The current study was approved by the Bioethics Commission from the University of Barcelona and by the Ethics Committee from the Vall d’Hebron University Hospital. Figure 1 provides details regarding the purpose of each stage of the study.

### 2.2. Stage One: Quantitative Methodology

#### 2.2.1. Participants

The study was conducted between May 2018 and July 2018 in three primary care centres with gastroenterology services located in Barcelona. Inclusion criteria involved the diagnosis of a gastrointestinal disorder. Gastroenterologists classified diagnoses into functional and organic. Patients were not included if they had an intellectual disability or cognitive impairment and did not have a sufficient level of Spanish or Catalan to understand the questions being asked.

#### 2.2.2. Instruments

##### Demographics

A questionnaire consisting of eight items related to gender, age, marital status, country of origin, educational qualifications, employment status, type of consultation (first or follow-up), and the reason for the consultation was provided to participants.

##### The Physical Functioning Subscale of the 36-Item Short-Form Health Survey

The physical functioning subscale from the SF-36 questionnaire was included to be able to assess the incongruence between gastroenterologists and patients with the same parameters as in former studies [17]. This subscale measures the person’s ability to perform different physical activities in an ascending effort gradient in a range from 0 to 100.

##### The Brief Symptom Inventory (BSI-18)

The BSI-18 [18], which is a shortened version of the Symptom Checklist-90 Revised Scale [19], is a brief psychological self-report symptom scale with a total of 18 items. Respondents are asked to indicate on a Likert scale ranging from 0 (not at all) to 4 (very much) to what extent they experienced symptoms in the last week. Symptoms are related to three dimensions: somatisation, depression, and anxiety (six items per dimension). A total score (general distress) was also used in the calculations. The Cronbach alpha of the instrument (α = 0.919) showed strong internal consistency.

##### The Patient-Doctor Relationship Questionnaire (PDRQ-9)

The PDRQ-9 is a brief measure that assesses the patient’s perception of their physician’s helping attitude [20]. This scale is composed of nine statements in which the patient is required to make a subjective assessment of the relationship that they have with their physician. Each response is chosen from a Likert scale of five categories ranging from 1 (totally disagree) to 5 (totally agree). The scores can range from a minimum of 9 to a maximum of 45, where a higher score indicates that the patient has a more positive view regarding the relationship they have with their doctor. The Cronbach alpha for the PDRQ-9 total score (α = 0.946) indicated strong internal consistency in our study.

##### Clinician-Rated Functional Impairment and Consultation Impressions

The Karnofsky Performance Status Scale (KPS) [21] was completed after the gastroenterologist had carried out the consultation with the patient. This instrument aims to measure the functional status of the patient. The questionnaire involves an 11-point scale that correlates to percentage values ranging from 100% (no evidence of the disease) to 0% (deceased). Additionally, a visual analogue scale (0–100) was used to assess the consultation impression.

#### 2.2.3. Determination of Incongruence between Clinicians’ Assessment and Patients’ Self-Reported Functionality

This study involved assessing the incongruence between the clinician’s assessment and the patients´ self-reported functionality. Using the same premise that was adopted in a previous study [17], patients were identified as having incongruent views with the gastroenterologist by using a value of 25 or greater on the difference between the KPS and the physical functioning subscale of the SF-36.

#### 2.2.4. Procedure

The first part of the study involved the administration of the sociodemographic questions. Patients were then asked to complete the physical functioning subscale of the SF-36, as well as the BSI-18. This first process was conducted before the consultation with the gastroenterologist.

Once they had seen the gastroenterologist, they were asked to complete the PDRQ-9. The three gastroenterologists that were involved in the study ensured that for each patient they saw, they recorded the KPS and the visual analogue scale values.

#### 2.2.5. Quantitative Data Analysis

We used two independent variables: incongruence (greater or less than 25) and diagnosis (functional or organic) in all analyses. Chi-squared and *t*-tests were used to compare the age, gender, educational level, marital, and employment status between those participants with greater or lower incongruence levels and those diagnosed with functional or organic disorders. A chi-squared test was used to analyse the relationship between the two independent variables. Subsequently, *t*-tests were used to compare the levels of distress between incongruence and diagnostic groups. Additionally, general linear models were used to calculate the interaction between incongruence and diagnosis using all study outcomes as dependent variables.

### 2.3. Stage Two: Qualitative Methodology

Once we verified the correlation between incongruence and psychological distress, as can be seen in the results section, the second stage of the research involved contacting patients from the first stage that had incongruent views with the gastroenterologist. Informed consent was obtained from each patient and physician, as sessions were recorded to be transcribed.

Eight focus groups with a total of 52 patients were carried out in three primary care units between September and October 2018. According to the recommendations of the ethics committees who authorised the study, no sociodemographic information was collected from these participants to ensure their confidentiality. The first part of the session involved asking questions related to their health status and daily life. The second half involved asking questions related to the health system.

A total of four individual physician interviews were conducted between October and November 2018. Interviews with physicians were conducted in order to obtain the perspective of the physicians regarding the healthcare reality of patients, as well as understand the implications that these disorders have on physicians working with this subgroup of patients. The interview involved six questions and lasted approximately an hour (see Table 1).

#### Qualitative Data Analysis

The focus groups and semi-structured interviews were carried out by a trained health psychologist and a PhD student with a background in clinical and health psychology (A.G.). They had also been part of the team in the quantitative section of the research. Both members received training from the principal investigator (F.J.E.-O.) before conducting the first focus group. Additionally, all semi-structured interviews were conducted by a trained therapist who was also provided with training and given a guide of the key questions that could be asked during the interview. All interactions, such as the instructions given to patients and the questions asked, were undertaken in Spanish.

All focus groups and interviews were transcribed verbatim. Once transcribed, we used the ATLAS.ti software to code and analyse the data [22]. A.G. then conducted thematic analysis by familiarising herself with the data and identifying common themes, patterns, and narratives through the analysis of words or fragments of text. The final themes were discussed with F.J.E.-O. and were reviewed and agreed upon by A.G.

## 3. Results

### 3.1. Stage One: Quantitative Results

#### 3.1.1. Sociodemographic and Psychosocial Characteristics

The approached sample was composed of 2261 patients. A total of 1562 patients met the inclusion criteria, were able to answer the questionnaires before the consultation, and were included in the analysis. The sociodemographic characteristics of the participants are presented in Table 2. Statistically significant differences were found between the incongruent and congruent groups for age, gender, educational level, and marital and employment status. Patients in the incongruent group were older than patients in the congruent group and were more likely to be female, less likely to have a higher educational qualification, to be married or in a stable relationship, and to be employed. Statistically significant differences between patients with a functional or organic diagnosis were found for age, sex, and employment. Patients with functional diagnoses were more likely to be younger, female, and employed. No statistical significance was found between incongruence and the type of diagnosis (*χ^2^* (1) = 0.131, *p =* 0.717).

#### 3.1.2. Distress and Patient-Physician Relationship

Independent samples *t*-tests were conducted to examine the differences across the incongruence and diagnosis groups with all outcomes (see Table 3). Statistically significant differences were found between incongruence and diagnosis groups for all distress dimensions. Patients in the incongruent and functional groups had, on average, higher levels of distress than patients in the congruent group. The *t*-tests yielded statistically significant differences in PDRQ scores between the diagnostic but not congruence groups. Lower average PDRQ scores were found among patients diagnosed with functional gastrointestinal disorders. Finally, clinicians’ impressions were statistically significantly lower for patients with incongruent views on their functionality, but no differences were detected by diagnosis. If corrections for multiple testing were applied, differences between patients with organic versus functional diagnoses in depression and PDRQ scores would not be considered statistically significant.

Table 4 shows the results of the general linear models using two independent variables: incongruence (i.e., congruent or incongruent) and diagnosis (i.e., functional or organic disorder); distress dimensions, PDRQ and clinician impression scores were used as dependent variables. As already observed in the *t*-tests, incongruence and the type of diagnosis had a statistically significant influence on each of the different dimensions of distress. However, no statistically significant interaction was found between incongruence and diagnosis for any of the distress subscales. As for the PDRQ, the results were reversed, with statistically significant results found only for the interaction, although with a very small effect size. Finally, the model carried out with clinical impressions showed statistical significance for incongruence and the interaction of the latter with diagnosis also with a very small effect size.

### 3.2. Stage Two: Qualitative Results

Table 5 and Table 6 provide a list of the main themes and the key patient and physician quotes (see tables in Appendix A for a full list of the patient and physician subthemes).

#### 3.2.1. Patient Focus Groups

##### Illness, Emotional, and Personal Problems

This theme referred to the patient’s physical discomfort and the emotional consequences derived from the digestive disorder. The results indicated that narratives about the consequences in their daily life were dominant in their discourse (25% of codes in this theme).

This was followed by stomach discomfort/stressful events (16%) and emotional consequences (12%). Patients that took part in the focus groups reiterated that having a gastrointestinal disorder affected their quality of life and that the onset or worsening of the disorder may be because of a stressful life event.

##### Disease-Health System Interaction

This theme referred to the interaction that took place between patients, healthcare system professionals, and the healthcare system itself. In other words, the codes from this theme were related to the functioning of the healthcare system and the global perspective that patients had about the system, the relationship they had with doctors, and the way in which these aspects affected their experiences when searching for a diagnosis and a cure. Twenty-five percent of the codes were related to explicit expressions of satisfaction with the care received. By contrast, nine percent of codes were about dissatisfaction. Additionally, patients showed feelings of uncertainty with waiting lists (15% of codes in this theme), as often there were long waiting periods to consult a doctor or to receive results, which had negative implications on their health status.

##### Health System

This theme referred to the patient’s opinion regarding the health system itself, the doctor’s attitudes towards the patients, and the way the doctors related to them. The patients considered the waiting lists (16% of codes in this theme) to be highly important and viewed them as being excessive. Patients also stated the importance of good communication with professionals (12%), and that despite the care they received, 11% of the codes reflected that the public health system is underfunded.

#### 3.2.2. Physician Interviews

##### Intervention

This theme referred to the characteristics and aspects of the intervention that the professionals found to be important when working with patients with gastrointestinal disorders. The participating professionals considered it essential to use a comprehensive intervention (16% of codes in this theme), for instance, by not limiting the intervention to a single specialty, which would help them understand the person as a whole. Thus, the ability to use a more comprehensive intervention would allow physicians to provide patients with a better explanation of their diagnosis.

##### Patients

This theme referred to the characteristics and attitudes of the patients that were relevant in the diagnostic and intervention process. According to the physicians interviewed, the psychological aspects (23% of codes in this theme) were crucial for patients with functional digestive disorders. The physicians observed that patients with FGIDs had a high prevalence of anxiety and depression, as well as difficulties with stress management and problem-solving.

##### Health System

This theme referred to the link between a lack of healthcare resources and lower quality of care. The specialists stated that there was a large volume of patients, and that the healthcare system was overloaded. Physicians identified that a requirement of any intervention with FGIDs was the development of a good doctor–patient relationship, which entailed a greater amount of time than any other part of the intervention. Lastly, participants expressed that they lacked mental health referral resources (25% of codes in this theme) and that they did not have an easily accessible resource for patients needing to be referred.

## 4. Discussion

The stage one objective was to examine the correlations of the type of digestive diagnosis and the congruence of the perception of severity and quality of life between gastroenterologists and their patients with psychological distress and perception of the help received experienced by the latter attended in primary care centres. The objective of the second stage of the study was to obtain more in-depth information from patients with incongruent views on their functioning with the gastroenterologist about their daily functioning and their attitudes towards the health system. In addition, the objective of this stage was to gain a better understanding from gastroenterologists regarding the healthcare implications of patients with digestive disorders and the effect this has on the work of gastroenterologists and other physicians working with this subgroup of patients.

In the first stage of the study, we found statistically significant differences between congruence and diagnostic groups in all the distress subscales, as the incongruent and FGID groups had higher distress scores. Thus, the findings supported our hypotheses that having incongruent views with the physician, as well as having a functional gastrointestinal disorder are related to higher levels of distress, with the former having a greater influence than the latter.

Interestingly, whilst patients with FGIDs were more likely to have lower levels of satisfaction with the care received, there was no significant difference between the congruent and incongruent groups. The situation was the opposite with the doctor’s impressions of the consultation which were statistically significantly lower for patients with incongruent views on their functionality. When investigating this further using focus groups with patients, it was clear that the patients were very satisfied with the service they were receiving from physicians, but that their dissatisfaction was attributed to the healthcare system and long waiting lists. Regarding the interviews with doctors, they reflected that there are psychosocial factors that could play an important role in the treatment but that there are different systemic impediments for a real transdisciplinary treatment. It can be observed that the mixed method approach provided us with a more holistic view and in-depth information regarding the gaps that still exist in primary healthcare, which can be used to help improve the service that patients receive.

Another interesting finding from this study was that no statistically significant interaction was found between incongruence and diagnosis. This indicated that these variables have a different relationship with distress. Thus, the presence of a functional diagnosis does not lead to higher levels of incongruence in the perception of functionality but rather different understandings of the illness may happen more frequently among patients with high levels of distress. These results are in line with previous research, as patients with FGIDs receiving specialised care had a similar psychological profile to patients with non-functional disorders [23]. Whilst similarities between the two groups were previously attributed to the setting in which the study took place (i.e., a specialised unit), it appeared that these similarities were also found in our study, which was conducted in primary care centres. Additionally, our results also mirrored findings from a study that found a significant link between somatisation, depression, and incongruence in a tertiary care setting [17,24]. Our findings were able to support the link further, as statistically significant results were found between incongruence and all the distress subscales.

Several limitations of the study should be considered. First, the cross-sectional nature of the study did not allow us to make inferences about causality. Second, all the instruments that were used in the study relied on self-reporting. Additionally, the assessment tool that was used to assess distress was not specific to digestive patients.

### Practice Implications

The present research helped to promote an equal platform for both patients and physicians. In doing so, this allowed us to more clearly identify the gaps that still exist in primary care for gastrointestinal patients and physicians. The implementation of detailed surveys, standardised patient assessments, and direct observations are needed to be able to provide actionable feedback to individual clinicians or health systems regarding how to achieve patient-centred care. Relevant stakeholders (i.e., patients) should be involved in order to capture important aspects of patient-centred care [25].

The current study provided patients with a ‘voice’, where their ideas could be transmitted back to physicians and the health system as a means of helping to improve their quality of life. Additionally, we aimed to provide physicians with the opportunity to give their concerns as a means of understanding how they can be better supported. Based on the feedback received from the focus groups and interviews, it is clear that certain aspects still need to be addressed.

## 5. Conclusions

In this study, we observed how incongruent views of functionality and type of digestive diagnosis contribute independently as markers of distress. We were able to understand that these patients perceived their functional status as being worse than what their physicians perceived due to multifactorial causes that ranged from their own perception of some aspects of the disease to their interaction with the health system. The ability for physicians to visualise a patient’s health status could provide them with a better understanding of the type of intervention that should be implemented and the specialities that should be involved in the process, including mental health professionals. Collaboration between specialties would help to provide a more accurate account of the patient’s quality of life, which may lead to more efficient care systems.

## Figures and Tables

**Figure 1 healthcare-11-00062-f001:**
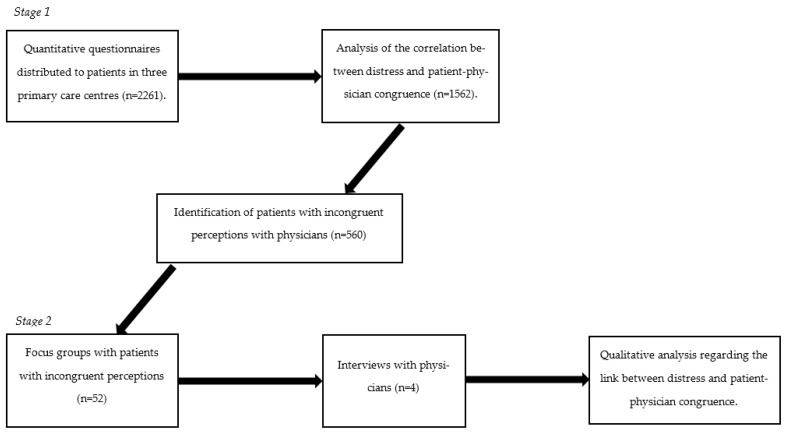
Flow diagram of the mixed methods research process.

**Table 1 healthcare-11-00062-t001:** A summary of the physicians’ interview questions.

Interview Questions
We have seen that your work involves working with patients with different types of disorders. What implications do the characteristics of functional, motor, inflammatory disorders have on your work?
Would you say that it is easier for you to make the diagnosis or develop an intervention with one type of patient than with another?
Do you explain the diagnosis differently according to the characteristics of the patient?
Have you noticed that patients with certain diagnoses are less able to accept their symptoms?
Depending on the characteristics of the patient, do you act differently to what you have noted down in the clinical history?
What benefits could a collaboration with mental health services bring you?

**Table 2 healthcare-11-00062-t002:** Sociodemographic and psychosocial characteristics of the sample by incongruence.

	Incongruent (n = 560)		Congruent (n = 1002)		Statistical Significance		Functional (n = 916)		Organic (n = 944)		Statistical Significance	
	**M**	**SD**	**M**	**SD**	** *t* **	** *p* **	**M**	**SD**	**M**	**SD**	** *t* **	** *p* **
Age (M ± SD)	64.86	14.82	52.22	17.26	15.225	<0.001	55.43	18.47	61.09	16.24	7.012	<0.001
	**N**	**%**	**N**	**%**	**O.R., 95% CI**	** *p* **	**N**	**%**	**N**	**%**	**O.R., 95% CI**	** *p* **
Sex (% females)	445	79.6	595	59.4	2.664, 2.092–3.391	<0.001	681	74.5	546	57.9	0.471, 0.386–0.573	<0.001
Education (% with a degree or higher)	53	9.5	220	22.0	2.698, 1.959–3.716	<0.001	148	18.6	128	16.1	1.118, 0.916–1.541	0.193
Marital status (% Married or stable relationship)	309	55.8	683	68.4	1.714, 1.383–2.123	<0.001	496	62.5	516	65.2	0.887, 0.723–1.089	0.252
Employment status (% employed)	108	19.3	483	48.3	3.901, 3.057–4.979	<0.001	322	40.4	279	35.2	1.246, 1.017–1.527	<0.05

**Table 3 healthcare-11-00062-t003:** Comparison of outcomes by congruence and diagnosis.

	Incongruent		Congruent		Statistical Significance		Effect Size	Functional		Organic		Statistical Significance		Effect Size
	M	SD	M	SD	*t*	*p*	d	M	SD	M	SD	*t*	*p*	d
Somatisation	4.62	4.20	8.49	5.73	13.904	<0.001	0.808	6.94	5.21	5.05	4.88	7.379	<0.001	0.374
Depression	3.46	4.58	7.08	6.22	11.896	<0.001	0.694	5.08	5.62	4.40	5.36	2.441	<0.05 *	0.124
Anxiety	3.54	3.84	6.14	5.46	9.864	<0.001	0.580	5.05	4.88	3.89	4.37	4.951	<0.001	0.251
General Distress	11.58	10.69	21.47	15.21	13.298	<0.001	0.793	16.96	13.68	13.23	12.80	5.487	<0.001	0.281
PDRQ	41.91	4.21	41.84	4.25	−0.305	0.761	−0.017	41.69	4.37	42.12	4.05	−1.970	<0.05 *	−0.103
Clinician impression	81.86	13.66	85.17	12.99	−4.671	<0.001	−0.250	83.63	13.99	83.70	14.09	−0.112	0.911	−0.005

* Differences that were not considered statistically significant if corrections for multiple testing were applied.

**Table 4 healthcare-11-00062-t004:** General linear models.

	Incongruence			Diagnosis			Interaction		
	F	*p*	ηp^2^	F	*p*	ηp^2^	F	*p*	ηp^2^
Somatisation	240.338	<0.001	0.135	57.776	<0.001	0.036	0.260	0.610	<0.0001
Depression	170.245	<0.001	0.100	5.693	<0.05	0.004	1.338	0.248	<0.001
Anxiety	120.520	<0.001	0.073	24.342	<0.001	0.016	0.078	0.780	<0.0001
General Distress	221.835	<0.001	0.129	31.826	<0.001	0.021	0.448	0.504	<0.0001
PDRQ	0.128	0.720	<0.0001	1.207	0.272	<0.001	4.382	<0.05	0.003
Clinician impression	22.483	<0.001	0.014	0.544	0.461	<0.0001	4.240	<0.05	0.003

**Table 5 healthcare-11-00062-t005:** Patient themes and quotes.

Patient Themes	Key Quotes
First theme (illness, emotional and personal problems)	‘If I had to go out, I did not eat so that I would not feel like going to the bathroom, if I had to go out to dinner with someone, I was already really nervous because well, I’m going to feel bad, I’m going to have to go to the bathroom’.
Second theme (disease–health system interaction)	‘What I think is that we have [some] magnificent, excellent doctors, I mean, I go very often to XXX because I have diverticula, I keep having flare-ups and I see that we have some great doctors.
Third theme (health system)	‘What I see wrong are the waiting lists, I had a CT scan in February and [only] now have they given me the results, they gave them to me in November’.

**Table 6 healthcare-11-00062-t006:** Physician themes and quotes.

Physician Themes	Key Quotes
First theme (intervention)	‘What you see of the psychological aspect of the patient is only one side, maybe you lose a lot of information that gets collected from other places, so in fact putting it together would be very useful’
Second theme (patients)	‘I believe that there are a number of patients who could benefit from psychological support, interventions, follow-ups and treatment for their functional disorder’.
Third theme (health system)	‘In the clinic, the truth is there isn’t much connection [with mental health services], and yes I would like to have more, for example, the referral is usually done by the head of the area, but the worst thing is that there is little feedback, and sometimes there are psychiatric patients, who for example, are already on different medication’.

## Data Availability

A database and calculation syntax can be downloaded as Appendix A.

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
