# Peer review of "Incongruent Views of Functioning between Patients and Gastroenterologists: A Mixed Methods Study"

_healthcare, 2022, doi:10.3390/healthcare11010062_

Round 1

Reviewer 1 Report

In this study, the authors evaluated the “Incongruent Views of Functioning between patients and Gastroenterologists using a Mixed-Method Enquiry”.  This study identified satisfaction and dissatisfaction that exist between the gastrointestinal patients and physicians. This study is relevant and interesting because it can help improve the services/attention the gastrointestinal patients can receive from physicians. Therefore the manuscript should be accepted after major corrections.

1. In the title -A Mixed-Methods Enquiry should be corrected to A Mixed-Methods Study

2. The literature report in the introduction is not adequate and should be improved.

3. In Figure 1, the authors should include the number of patients in each box for easy flow and understanding. The Figure is not relevant without specifying the exact number of patients.

4. The results of demographic study was not properly presented, the total number of female and male patients should be specified.

5. The authors should specify the age grade/range (for instance 11-20, 21-30 years) and gender with the highest and lowest levels of satisfaction and dissatisfaction among the examined patients.

6. The authors should discuss how educational levels of gastrointestinal patients influence or affect the outcome of this study.

7. Page 4 line 2-3 Patients in the incongruent group were older than patients in the congruent group and were more likely to be female. This statement “were more likely to be female” indicates that the authors are not sure of the gender.

8. In page 5 In general patients are highly satisfied with professionals (25%). I do not think that 25% should be classified as highly satisfied.

9. In tables 2 and 3 with statistical analysis, the authors should define the letters meaning.

10. English language needs revision (few typographical and grammatical errors were detected).

Author Response

Dear Reviewer,

We would like to resubmit the manuscript retitled “Incongruent Views of Functioning between Patients and Gastroenterologists: A Mixed-Methods Study”. We have tried to address all the issues raised and have accommodated and incorporated all the necessary revisions. Below you can find all the points raised and our respective annotation to the modification.

First, we have uploaded a clean version of the manuscript including tables and figures, followed by a tracked changes version of the modified documents in the Editorial Manager application.

We look forward to your comments.

Sincerely,

The authors

Reviewer 2 Report

A very practical study especially dealing with complex multi domain disorders of FGID

The authors have concluded aptly but the the description of results and the discussion needs to be simplified to ensure reader interest 

The discussion needs to be enhanced 

Author Response

(The authors gave the same response as above.)
